# CCT: Cross-consistency training for Clone Detection and Code Search Tasks

## Abstract

Clone detection is a well known task, which could be formulated on any programming language. Although to the best of our knowledge there is no cross-lingual clone detection task formulated. In this work we formulate such a task alongside with a specific training procedure CCT for a deep leaning language model. This procedure allows CCT-trained model to outperform the existing approaches on POJ-104 benchmark with result of 95.67% MAP and on newly created cross-lingual clone detection benchmark XCD. Moreover, CCT model shows new state of the art results in code search task AdvTest 47.15% MRR.

## 1 Introduction

In software development practice in it is sometimes important to identify the code with the same effective output. This could be useful e.g. for unification and control of side effects. To meet this need there was clone detection task formulated (Mou et al., 2016). In the mentioned work the task is formulated for `C/C++` programming code. Although for compiled languages it seems to be the most profitable to *detect* similar behaviour of the code, instead of compiling and running, it still could be useful for many other languages, including interpreted ones. The next step which could be done, one could detect the same output for the programming code in different languages[1]. We formulate cross-lingual clone detection task and establish some baselines on it.

There are various approaches to solve clone detection task problem starting from algorithmic based methods (**?**) and continuing with modern machine learning ones (). Most machine learning approaches are based on embedding representation of the code snippet. This approach allows to find duplicate code snippets by similarity between their embedding representation. The performance of such systems depends on the quality of obtained embeddings. We present a novel technique of training (CCT) for language models which allows them to embed the code snippets in effective way. We demonstrate this on previously formulated clone detection task POJ-104 (Mou et al., 2016) and on newly formulated cross-lingual clone detection task XCD.

Interestingly, we also found out that this CCT technique allows a model to produce representations useful for code search task also. Code search itself is a task where a code snippet should be mapped to some text description, as formulated in (Lu et al., 2021a).

The contributions of our work is as follows: (i) we present a pre-training method CCT allowing a model to align code snippets in different languages; (ii) we present a novel cross-lingual clone detection task XCD; (iii) we present results for a language model trained with CCT on clone detection tasks POJ-104 and XCD; (iv) we present the results of CCT-model for code search AdvTest task.[2]

## 2 Datasets

In our work we use two types of the datasets, one is for clone detection, the other is for code search.

---

[1] As an example of the programmes with the same effective output one could refer to `http://helloworldcollection.de/` website, which contains "Hello, world!" snippets in 603 programming languages.

[2] We are going to release CCT code after the review process is over.

## Cross-Lingual Alignment

Figure 1: Here is a difference between strong and weak cross-lingual alignment. In a strongly aligned embedding space, the most semantically similar items are always the closest, regardless of language. A weakly aligned multilingual embedding space just enables zero-shot transfer between languages.

### 2.1 CODE SEARCH

For code search we use CodeSearchNet dataset first described in (Husain et al., 2019a). The original version of this dataset consists of natural languages queries paired with most relevant code snippets written on six programming languages. Each snippet is a code function collected from GitHub open source code. (Lu et al., 2021a) modified CodeSearchNet and created AdvTest dataset.

**CodeSearchNet AdvTest** is a Python language only dataset constructed from the CodeSearchNet corpus. Each example includes a function paired with a document. The authors of AdvTest followed the original work (Husain et al., 2019a) in taking the first paragraph of the documentation as the query for the corresponding function. To improve the quality of the dataset, they filter it by removing the following examples:

- Examples whose code could not be parsed into abstract syntax tree.

- Examples whose document is shorter than 3 or larger than 256 tokens.

- Examples whose document contains special tokens such as "http://".

- Examples whose document is empty or not written in English.

The filtered dataset contains 251,820 / 9,604 / 19,210 examples for training/validation/testing respectively. The idea of this dataset is based on the following observation. After normalizing function or variable names with special tokens, the authors observe that the Mean Reciprocal Rank (MRR) scores of RoBERTa (**?**) and CodeBERT (**?**) for the code search task on original CodesearchNet (Husain et al., 2019a) dataset drop from 0.809 to 0.419 and from 0.869 to 0.507, respectively, in Python programming language. To better test the understanding and generalization abilities of the model, they normalize function and variable names in testing and development sets like $func$ for the function name and $arg_i$ for the i-th variable name. The task aims to search source codes from candidates for a natural language query. In contrast to the testing phase of previous works (Husain et al., 2019a; **?**) that only involved 1,000 candidates, the authors use the entire testing set for each query, which makes **CodeSearchNet AdvTest** dataset more difficult. The training set for this task comes from the filtered CodeSearchNet dataset (Husain et al., 2019a).

As a metric for AdvTest MRR is used. It is defined as:

$$\text{MRR@20} = \frac{1}{Q}\sum_{i=1}^{Q}\frac{1}{rank_i},$$

where $Q$ is set queries and $rank$ is a position of ground truth answer document among the ranked candidates.

## 2.2 CLONE DETECTION

The task aims to retrieve semantically similar codes given a code as the query and we use POJ-104 dataset to perform it. **POJ-104** dataset described in (Mou et al., 2016) comes from a pedagogical programming open judge (OJ) system that automatically judges the validity of submitted source code for specific problems by running the code. We use the POJ-104 dataset, which consists of 104 problems and includes 500 student-written C/C++ programs for each problem. The task of POJ-104 aims to retrieve other programs that solve the same problem given a program. The problems are grouped in three sets (64/16/24) for training, validation, and testing respectively.

The metric for POJ-104 dataset is Mean Average Precision (MAP). We start with defining average precision (AP).

$$\text{AP} = \sum_{i=1}^{100}(R_i - R_{i-1}) \cdot P_i,$$

where $R_i$ and $P_i$ are the precision and recall at the $i$ threshold, i.e. they are computed taking into account only top $i$ items from the candidate list. MAP is the mean of AP over all the queries. It is important to mention that for POJ-104 the maximal possible $i$ is 499, since there is only 500 candidates in total.

## 2.3 CROSS-LINGUAL CLONE DETECTION

In previous works, cross-lingual and multilingual abilities of CodeLM were not sufficiently investigated. To fill this gap, we are introducing a new Cross-lingual Clone detection/Code retrieval dataset (**XCD**). Dataset is constructed in three different setups: in similar to BUCC dataset (Xu et al., 2018) retrieval way, in similar to code clone detection way POJ-104 (Mou et al., 2016) and in a hybrid way. Moreover, we investigated zero-shot transfer from python to Java, Ruby, PHP, Go and JavaScript on CodeSearchNet dataset for previously introduced CodeLM models and our $CCT - LM$.

As a data source CodeForces submissions dump was leveraged. We filtered 100 problems, which were not used during pre-training. For similar to BUCC approach, we randomly choose 3 accepted solutions per problem on 9 different program language. Positive examples are accepted solutions for same problems in different language. In total, for each language in the corpus we mined *50000* randomly chosen submissions, and 10000 of them have a positive pair. The dataset was split dev and test 20/80 respectively.

For *code clone retrieval* approach we follow (Hu et al., 2020; Tien & Steinert-Threlkeld, 2022), where the models are evaluated on the dev and test split of the corpora and the threshold of the similarity score cutting off translations from non-translations is optimized for each language pair. The similarity scores are calculated based on dot products of CLS vectors representations, also in pre-training. Our task is interpreted as binary classification, thus classic F1 is used.

For *Clone detection* approach, we follow POJ-104 dataset design. For 100 problems were mined, 100 accepted solutions per problem on 9 different program languages. The task aims to retrieve 100 snippets per language solving same problem from the test set in zero-shot setup.

Finally, for a hybrid approach, we leverage data from first dataset and retrieval method from the second. As a metric we chose MRR@20. It is the same metric as in AdvTest.

## 2.4 ADDITIONAL MARKUP

In addition to the solution status ("Accepted" or not), we also mined the statuses of errors in the solutions, since the platforms used for problem solving are often provide them. Finally, we mined more than 97 million code snippets in more than 10 programming languages.

### 2.4.1 VERDICTS

CodeForces platform can return 15 types of verdicts for submitted solution. We split verdicts on 4 groups: Defect (code marked as contained defect), Skip (code which cannot be judged), Accepted (code without any detected defect), Wrong solution (code failing some tests respectively). Below we describe the most common of verdicts:

**Memory limit exceeded** The program tries to consume more memory than is indicated in the problem statement. (Wrong solution)

**Time limit exceeded** The program hadn't terminated in time indicated in the problem statement. (Wrong solution)

**Runtime error** The program terminated with a non-zero return code (possible reasons: array out of bound error, division by zero, stack overflow, incorrect pointers usage, etc.) (Defect)

**Wrong answer** Wrong answer. (Wrong solution)

**Idleness limit exceeded** The program did not use the CPU time for considerable time. (Defect)

**Denial of judgement**, **Judgement failed** The solution was impossible to run, perhaps, due to a judging error. The most probable cause is an error in the program (for example, using extra large arrays). (Defect)

**Rejected on** The program does not pass tests and we do not know why is happened. (Skip)

**Accepted** Solution passed all tests. (Accepted)

## 3 METHOD

In this section, we will describe our pre-training approach (**CCT**). The goal of our method is to robustly learn the embedding space of code snippets and create strong alignment between code solving the same problems across different programming languages. To achieve this, we introduce a new contrastive learning objective: for a randomly masked code snippet (extracted from XCD in our experiments), we train the CLS vector such that the CLS embedding will be closer to the snippet in another language or user but solving the same problem or the same snippet but masked differently than a random or similar but different (hard negative) snippet. More formally, for a given random problem from CodeForces we have $n$ relevant snippet $D = [d_1, d_2, ..., d_n]$ in different languages from $L = [l_1, l_2, ..., l_n]$ and $j \in \overline{1...n}$ with $p(j) = \frac{1}{|D|}$ . We optimize noise contrastive estimation loss:

$$\mathcal{L}_{contr} = \sum_{i=1}^{n} -\log \frac{e^{\text{sim}(d_i, d_j)}}{\sum_{k=1}^{m} e^{\text{sim}(d_j, d_k)} + e^{\text{sim}(d_i, d_j)}}, \tag{1}$$

So in general, code solving the same problem should have similar representations and the ones with different problems should have different representations.

Also in final loss function we specified another two: error detection loss $\mathcal{L}_{bug}$ and standard language modelling loss $\mathcal{L}_{lm}$. Error detection loss is native for code data and can help model to deeply understand code syntax and recognise errors. Task is formulated such as binary classification of code which can raise error during code execution. The language modelling loss is a standard masked token prediction loss, which is widely used for language models, including the models trained on programming code.

Our final loss function composes all the listed ones: $\mathcal{L} = \mathcal{L}_{contr} + \mathcal{L}_{bug} + \mathcal{L}_{lm}$.

### 3.1 HARD NEGATIVE MINING

Previous works in contrastive learning (Qu et al., 2021; Izacard & Grave, 2020) show the importance of hard negatives examples. All of them are using iterative training to get the ones. However, our data already contain some strong negative examples, namely, solutions solving the same problem from same users but failing the tests. Thus, our hard negative examples are mined as a failed solution from the same user if it exists. If there is no such solution, then a failed solution from a random user is used. If there is no such solutions, then we use simply a random submission.

## 4 EXPERIMENTS

In this section, we will describe details about parallel data pre-training and pipeline for fine-tuning for cross-lingual open domain question answering and cross-lingual sentence retrieval tasks.

### 4.1 PRE-TRAINING

Generally, we initialize our model with pre-trained GraphCodeBERT$_{base}$ (Guo et al., 2020). We call these models CC-LM$_{base}$. We use an AdamW optimizer with a learning rate 1e-4, weight decay of 0.01, and linear learning rate decay. We also use gradient cashing for pre-training, similar to (Gao & Callan, 2021). The accumulated batch size was equal to 500. We train our models on 4 NVIDIA Tesla V100 GPUs.

### 4.2 RESULTS

| Metric | Clone Detection MAP | Code Search MRR |
|---|---|---|
| RoBERTa-base (Liu et al., 2019) | 76.67 | 18.33 |
| CodeBERT (Feng et al., 2020b) | 82.67 | 27.19 |
| SynCoBERT (Wang et al., 2021a) | 88.24 | 38.1 |
| CodeRoBERTa | - | 42.35 |
| GraphCodeBERT (Guo et al., 2020) | 85.16 | - |
| CasCode (Gotmare et al., 2021a) | - | 43.98 |
| (Villmow et al., 2022) | 91.34 | - |
| CCT-LM$_{base}$ | **96.73** | **47.18** |

Table 1: Results on code clone detection on POJ-104 and Code Search on AdvTest

In Tab. 1 we present the results of our CCT-LM model (a model pre-trained with proposed CCT method) in comparison to the existing approaches. As one can see our CCT-LM outperforms all the previous models by a large margin.

| | python | java | csharp | ruby | js | haskell | php | ocaml | perl | Avg |
|---|---|---|---|---|---|---|---|---|---|---|
| | | | | | F1 | | | | | |
| graphcodebert-base | 0.02 | 0.05 | 0.00 | 0.04 | 0.00 | 0.02 | 0.01 | 0.03 | 0.01 | 0.02 |
| graphcodebert-base-POJ-104 | 0.04 | 0.00 | 0.01 | 0.06 | 0.07 | 0.08 | 0.06 | 0.06 | 0.06 | 0.05 |
| CCT-LM$_{base}$ | **22.24** | **18.39** | **17.33** | **23.33** | **10.46** | **17.64** | **21.43** | **17.01** | **16.40** | **18.24** |

Table 2: Performance on cross-lingual code retrieval dataset.

#### 4.2.1 CROSS-LINGUAL RESULTS

Results for cross-lingual code retrieval task are presented in Tab. 2. As it can be seen, task is the more complex than similar task for Natural language, as a reference we can use (Sorokin et al., 2022). It is interesting, that we cannot track knowledge transferring from POJ-104 dataset as the metrics are low. However, CCT approach shows better result due to more obvious cross-lingual pre-training.

| | python | java | csharp | ruby | js | haskell | php | ocaml | perl | Avg |
|---|---|---|---|---|---|---|---|---|---|---|
| | | | | | MAP@100 | | | | | |
| graphcodebert-base | 7.21 | 9.25 | 1.33 | 4.28 | 1.59 | 5.78 | 6.08 | 2.90 | 10.37 | 5.42 |
| CCT$_{LM}$ | **87.42** | **55.99** | **65.35** | **72.12** | **74.32** | **81.05** | **83.21** | **71.53** | **71.89** | **73.65** |

Table 3: Performance on clone detection cross-lingual dataset.

For clone detection setup the results are presented in Tab. 3. Similarly to POJ-104 we leverage MAP@100 since for every example we have 100 positive duplicates. As one can see, our method is strongly outperforms the baseline.

| | | | | | MRR@20 | | | | | |
|---|---|---|---|---|---|---|---|---|---|---|
| | **python** | **java** | **csharp** | **ruby** | **js** | **haskell** | **php** | **ocaml** | **perl** | **Avg** |
| BM25 | 0.00 | 0.00 | 0.00 | 0.00 | 0.00 | 0.00 | 0.00 | 0.00 | 0.00 | 0.00 |
| graphcodebert-base | 2.08 | 5.42 | 0.22 | 2.59 | 0.80 | 1.99 | 2.90 | 1.40 | 5.23 | 2.51 |
| graphcodebert-base-POJ-104 | 27.10 | 20.04 | 19.44 | 30.98 | 28.37 | 19.70 | 32.89 | 30.08 | 39.98 | 27.62 |
| $CCT_{LM}$ | **74.97** | **62.08** | **58.77** | **80.60** | **74.56** | **62.27** | **81.21** | **72.64** | **79.16** | **71.80** |

Table 4: Performance on hybrid cross-lingual dataset.

Results for hybrid seen more interpreted 4. Ability of Code language models to transfer knowledge between programming language more obvious, since metrics improvement is noticeably greater between fully unsupervised method and pre-training on monolingual clone detection. Moreover, BM25 (a strong baseline for natural language information retrieval tasks) does not work for cross-lingual code retrieval task. That shows ability of CodeLM models to cross-lingual semantic search.

## 5 ANALYSIS

Fig. 2a shows the abstract representation of the idea that semantically aligned embedding space is language-agnostic. On Fig. 2b there are the actual embeddings for samples in six languages before and after CCT training. As one can see, CCT representations of code snippets are not aligned by the same language. Thus we can conclude that CCT training significantly improves the semantic closeness for different language code snippets. The code snippets here are solutions for sampled 12 tasks from CodeForces on different programming languages.

### 5.1 ABLATION STUDY

| Metric | Clone Detection
**MAP** | Code Search
**MRR** | Defect-detection
**Accuracy** |
|---|---|---|---|
| GraphCodeBERT | 85.16 | 45.80 | 62.51 |
| GraphCodeBERT + $\mathcal{L}_{contr}$ | 95.92 | 29.93 | 61.05 |
| GraphCodeBERT + $\mathcal{L}_{contr}$ + $\mathcal{L}_{LM}$ | 95.67 | 47.18 | 63.68 |
| GraphCodeBERT + $\mathcal{L}$ | 96.03 | 45.22 | 64.91 |
| GraphCodeBERT + $\mathcal{L}$ + s.l. | 96.46 | 47.33 | - |
| GraphCodeBERT + $\mathcal{L}$ + s.l. + v.f. | 96.73 | 47.57 | 65.58 |

Table 5: Results on code clone detection on POJ-104 and Code Search on Adv Test; s.l. stands for size limitation, v.f. - verdict verification.

In this section, we will discuss the effectiveness of the proposed approach. Tab. 5 displays the result of retrieval on clone detection, Code search, and Defect detection tasks. We decided to add Defect detection task for our analysis, since a part of our loss is easily aligned with this task. As a dataset for it we used the one presented in (Zhou et al., 2019). This dataset contains 21,854/2,732/2,732 for train/dev/test split. As a metric it uses straightforward Accuracy.

We compare here the GraphCodeBERT base model with different pre-training loss, verdict filtration, this size limitation. How can be seen, changes gives not uniform metrics gain. Pure contrastive loss significantly reduce quality on all tasks but significantly improve MAP on Clone Detection task. Adding masked language modelling loss fill in these metrics gap, probably due to the representations from the general pre-training are saved better. The biggest increase for Code Search task, can be explained by similarity of CF data and functions from Code search data. $\mathcal{L}_{bug}$ improve quality for significantly for Defect detection task and a little bit for the clone detection one. Limitation of size significantly reduce number of code snippets and problems respectively. However, it improves metrics for all tasks. Same result for verdict filtration.

Generally, the increase from the change is uneven and it is possible to achieve a better result in other configurations.

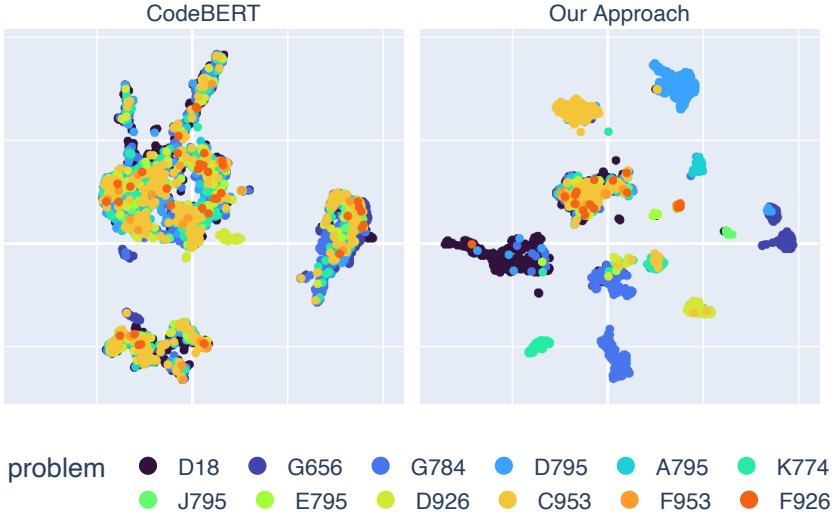

(a) Projected embeddings of sampled 12 coding problems.

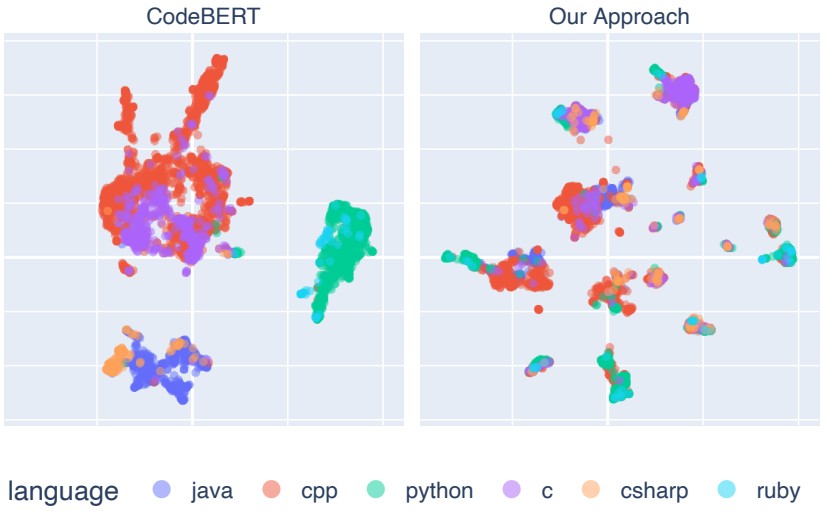

(b) Same embeddings differentiated by programming language.

Figure 2: Analysis of same-language bias in the cross-lingual embedding space.

| | F1 | | | | | |
|---|---|---|---|---|---|---|
| | **java** | **ruby** | **php** | **go** | **js** | **Avg** |
| CodeBert$_{base}$ | 46.37 | 50.65 | 37.83 | 50.65 | 50.48 | 47.19 |
| GraphCodeBert$_{base}$ | 47.33 | 59.95 | 37.47 | 60.28 | **52.04** | 51.41 |
| CCT-LM$_{base}$ | **48.71** | **62.25** | **42.78** | **61.44** | 51.06 | **53.24** |

Table 6: Zero-shot retrieval on CodeSearchNet dataset.

It is interesting to analyse the zero-shot results presented at Tab. 6. As one can see, the existing approaches are showing pretty good results, even though they have not been pre-trained on retrieval

task. It is an additional evidence of the pre-trained language models' power. Although, our method is leveraging its cross-lingual ability and improves over the baselines in zero-shot setup. However, for JavaScript (js) language our pre-training method CCT shows lower performance than the baseline, we leave this fact for further investigation.

## 6  RELATED WORK

Our methods are inspired by natural language processing field, thus we present some works from this field in addition to the programming language processing ones.

**Datasets**   To the best of our knowledge, the first attempt to make a code search dataset from natural language question-answer pairs was described in (Li et al., 2019). The resulting dataset NeuralCodeSearch contains about 4 million code snippets scraped from GitHub and more importantly an evaluation part that was curated from a dump of StackOverflow questions. This part contains only 287 question-snippet pairs. The next approach is described in work (Husain et al., 2019b). This dataset CodeSearchNet is also built from a GitHub dump, but in this case, the authors split the method bodies into the code itself and a description. This dataset contains 2 million code snippet & description pairs in 6 programming languages including Python. This dataset is partially used in the next work (Hasan et al., 2021). In this paper, the authors take several datasets – CodeSearchNet and 3 others and created a bigger one. From CodeSearchNet they used the Java part and Python part which is translated automatically into Java. The resulting dataset contains 4 million code snippet-description pairs. Interestingly, the biggest dataset was built in the earlier work (Gu et al., 2018). This dataset, CODEnn-Train, contains 18 million method definition & one sentence description pairs and consists of code written in Java. More recent work CodeXGLUE presented in (Lu et al., 2021b). It is a machine learning benchmark collection of datasets for code understanding and generation tasks, which includes a modification of mentioned CodeSearchNet. CodeXGLUE provides a benchmark for different code-to-code, code-to-text, text-to-code tasks, and code search as one of them. This benchmark includes code in 10 programming languages. Another multi-task dataset was presented in (Puri et al., 2021). It has 14 million code snippets in 5 programming languages.

For clone detection there are two main works POJ-104 (Mou et al., 2016) and BigCloneBench (Wang et al., 2020). The first one represents a comparatively small corpus of C++ problem solution from a student judging system. While the other is bigger, and includes a lot of mined data in several langauges.

**Code Search Approaches**   There is a line of early work on code search (Bacchelli et al., 2010; Brandt et al., 2010; Campbell & Treude, 2017; Chan et al., 2012). These works mostly relied on classic information retrieval, which turned out to be still a strong baseline in our experiments. (Brandt et al., 2010; Barzilay et al., 2013; Ponzanelli et al., 2014) were relied on the existing industry scale web search engines.

In (Gu et al., 2018) modern dense vector representations were used for information retrieval. The authors use two recurrent neural networks to represent the code and textual descriptions. In the work (Feng et al., 2020a) the authors presented a language model-based approach to produce these representations. The authors of paper (Gotmare et al., 2021b) use three Transformer-based models, 2 as the encoders and 1 as a classifier, for a hierarchical representation of code and text, although they propose to share the parameters in the encoders, it lowers the final quality of their model. Our model in contrast uses a single encoder for embedding the queries and documents and skips the classifier part.

There is also a line of recent work, which address different code search aspects, not directly the code search itself in the formulation of CodeSearchNet (Chen & Abedjan, 2021; Hammad et al., 2022; Gu et al., 2022; Di Grazia, 2022; Luong et al., 2022; Zhang et al., 2022).

**Code LM**   After the success of BERT-like models for natural language, the attention of the community was pointed to programming languages. Thus recently there were presented several pre-trained programming language models, namely CodeBERT (Feng et al., 2020a) which is a bimodal pre-trained model for programming language and natural language, based on RoBERTa (**?**) transformer architecture, trained on masked language modelling and replaced token detection objectives; Graph-

CodeBERT (Guo et al., 2021) model uses data flow in the pre-training stage to solve MLM, edge prediction and node alignment tasks; SynCoBERT (Wang et al., 2021b) model uses multi-modal contrastive learning to achieve better code representations. The model is pre-trained on identifier prediction and abstract syntax tree edges prediction tasks.

**Natural Language Systems** Recent research was focused on creating non-English question answering datasets and applying cross-lingual transfer learning techniques, from English to other languages. Until recently, the availability of appropriate train and test datasets has been a key factor in the development of the field: however, in recent years, many works have focused on the collection of loosely aligned data obtained through automatic translation or by parsing similar multilingual sources. (Lee & Lee, 2019) have shown transfer learning applicability for cross-lingual QA with training on English data and evaluation on Chinese data. (Artetxe et al., 2020) studied cross-lingual transferability of monolingual representations of a transformer-based masked language model. (M'hamdi et al., 2021) examined a cross-lingual optimization-based meta-learning approach (meta-training from the source language to the target language(s) + meta-adaptation on the same target language(s) for more language-specific adaptation), to learn to adapt to new languages for question answering. (Gao & Callan, 2021) proposed unsupervised pre-training for dense passage retrieval, although the authors concentrated on retrieval itself, ignoring cross-lingual nature of the data.

In most previous approaches the authors use extractive models to generate the actual answer. This could be explained by the mental inertia from SQuAD-like datasets. By SQuAD-like we mean a dataset where labelled data includes an explicitly stated question, a passage, containing an answer, and a span markup for the answer. Such markup was presented for the question answering task called SQuAD in (Rajpurkar et al., 2016). Recently several works on cross-lingual generation of answers from raw texts has been presented. Kumar et al. (2019); Chi et al. (2019) studied cross-lingual question generation. Riabi et al. (2020) also suggested a method to produce synthetic questions in a cross-lingual way, using Multilingual MiniLM. Shakeri et al. (2020) proposed a method to generate multilingual question and answer pairs by a generative model (namely, a fine-tuned multilingual T5 model), it is based on automatically translated samples from English to the target domain.

Generative question answering was mostly considered in previous work for long answers datasets. However, FiD model (Izacard & Grave, 2021) archives competitive results on SQuAD-like datasets, where an answer is supposed to be short text span. For open domain question answering, one of the first approaches named RAG used generative models was presented in (Lewis et al., 2021). A key idea of this RAG model is to process several (top k) passages from the retriever in the encoder simultaneously. The produced dense representations of the passages are used in the decoder for the answer generation, this process is called fusion. Processing the passages independently in the encoder allows a model to scale to many contexts, as it only runs self-attention over one context at a time.

For question answering over knowledge graph, (Zhou et al., 2021) studied unsupervised bilingual lexicon induction for zero-shot cross-lingual transfer for multilingual question answering, in order to map training questions in the source language into those in the target language as augmented training data, which is important for zero-resource languages.

## 7    CONCLUSION

The understanding of semantic similarity is an important ability to solve many different tasks. We present a new method to improve this ability and demonstrate its improvement on tasks of Clone Detection and Code Search. We also formulate a novel task of Cross-lingual Clone Detection, thus we present XCD dataset and describe three different setups on it, namely clone detection, clone retrieval, and a hybrid one. Our model had outperformed strong baselines in all presented tasks, proving that our pre-training method gives mode power for semantic similarity understanding to a particular language model.

We hope that our method will be helpful in other programming language processing tasks, which we have not covered in our work. So we leave them as a future work. But in addition, we think that method could be useful for other researchers in this and other fields.

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
