# OpenReview forum: "CCT: Cross-consistency training for Clone Detection and Code Search Tasks"
_ICLR.cc/2023/Conference — Submitted to ICLR 2023_

### Official Review · Reviewer_Fdtp · 2022-10-19

**Confidence:** 3
**Correctness:** 3
**Technical Novelty And Significance:** 2
**Empirical Novelty And Significance:** 2
**Recommendation:** 5

**Clarity, Quality, Novelty And Reproducibility:**

The paper is mostly clear with focused contributions. The motivation can be made clearer and the presentation of the paper can be more reader-friendly. Missing these parts, I would judge that the work has borderline originality, considering both the contribution in dataset and the pretraining objective that the paper proposes.

The paper should be reproducible; the authors have detailed their technology as well as the dataset used in the paper. Although it would be great if the authors can also release the model.

**Strength And Weaknesses:**

Strength:
- The paper extends the search/detection tasks on programming languages, notably from single language setting to a multi-lingual setting; proposing the XCD dataset in the paper.
- For tasks related to programming language search/retrieval, the paper propose a new pretraining objective. It shows that the new pretrained model outperforms previous models in general with associated ablation studies.

Weakness:
- I miss the motivation to study the multi programming language settings. I encourage the authors to think through the impact that such settings might bring from research to application perspectives and explain to readers why studying these are important.
- The presentation of the paper is not satisfactory: 1) in related works, the authors specify natural language systems in detail, I can guess why multilinguality and the development there connects to the work in this paper; but they are not explicit and confuse readers like me. 2) 3.1 mentions hard negative mining which is interesting, however, I fail to see its impact in the result section. 3) CCT is mentioned in the abstract without readers knowing what it means 4) in several places, the reference shows a ? mark simply.


**Summary Of The Paper:**

The paper studies some tasks related to programming language (e.g. clone detection, code search). The paper makes two contributions in the domain: firstly, through online OJ code mining, the paper proposes a new benchmark XCD that evaluates model over multiple program languages. Secondly, the paper proposes a new pretraining approach called CCT which combines masked language modelling with contrastive loss as well as code specific bug loss. Empirical results show that the proposed model achieves good performance compared to previous approaches in various tasks including code retrieval and code cloning as well as in various settings including single programming language settings, multi-language settings and zero shot setting.

**Summary Of The Review:**

The paper studies some tasks related to programming language (e.g. clone detection, code search). The paper makes two contributions in the domain: firstly, through online OJ code mining, the paper proposes a new benchmark XCD that evaluates model over multiple program languages. Secondly, the paper proposes a new pretraining approach called CCT which combines masked language modelling with contrastive loss as well as code specific bug loss. The paper has done experiments showing convincing results.

On the other side, the paper misses a strong motivation explaining why this line of work can be impactful and interesting. Although with clear contributions, the discussion of the paper (e.g. natural language system related works) is confusing; there is some serious formatting issues as well in the paper.

---

### Official Review · Reviewer_Y4fF · 2022-10-22

**Confidence:** 4
**Correctness:** 2
**Technical Novelty And Significance:** 2
**Empirical Novelty And Significance:** 2
**Recommendation:** 3

**Clarity, Quality, Novelty And Reproducibility:**

The significance of the work is weak. The paper needs to explain why to formulate the cross-lingual clone detection task. Can it be applied to some real scenarios? How is it compared with existing approaches?

**Strength And Weaknesses:**

Strengths:

(+) Code search and clone detection are important and useful for software development and maintenance.

(+) The authors discuss data quality issues and evaluate the approaches on a good quality dataset.

Weaknesses:

(-) The significance of the work is weak.

(-) Experiments are weak.

(-) The analysis of the experimental results is not sufficient.

(-) The presentation of the paper is poor.

**Summary Of The Paper:**

This paper targets cross-lingual clone detection. It presents a contrastive learning-based technique to learn the embedding space of code snippets and to create alignments between code solving the same problems across different programming languages. The authors also conduct experiments on code search and code clone detection. The experimental results show the effectiveness of the proposed model.

**Summary Of The Review:**

The experimental settings are weak: important baselines [1,2] are missing and the comparisons are unfair. Specifically, previous work [3] has found that the larger batch size can boost the performance of contrastive learning-based models. The proposed methods are contrastive learning-based and batch size is set to 500 in this paper.  However, the baselines are all smaller than 500. It is better to study the effect/contribution of large batch sizes.

[1] Guo, D., Lu, S., Duan, N., Wang, Y., Zhou, M., & Yin, J. (2022). UniXcoder: Unified Cross-Modal Pre-training for Code Representation. ACL 2022.

[2] Zhang, J., Wang, X., et al., A novel neural source code representation based on abstract syntax tree. In IEEE/ACM 41st International Conference on Software Engineering (ICSE 2019), May 2019.

[3] Shi, E., Gub, W., Wang, Y., et al., (2022). Enhancing Semantic Code Search with Multimodal Contrastive Learning and Soft Data Augmentation. arXiv preprint arXiv:2204.03293.

The analysis of the experimental results is not sufficient. This paper does not provide a convincing argument why the proposed methods is effective. In which cases can it perform better than current baselines and why? Specifically, in the section EXPERIMENTS, the authors only present the experimental results on the table and conclude that the proposed methods outperform all the baselines. However, first, it is better to explain what the baselines are. For example, what are the meaning of “graphcodebert-base” and “graphcodebert-base-POJ-104” in Table2? What is the difference between the “GraphCodeBERT” in table1 and “graphcodebert-base” in table 2. Second, the paper needs to provide some convincing arguments to justify why the proposed approach is effective.

The presentation of the paper is poor. Important details and references are missing or hard to understand. The figures/tables miss some explanations. In addition, there are some grammar errors in the paper:

• In software development practice in it is sometimes important to identify the code with the same effective output. (Should delete “in”).

• Ability of Code language models to transfer knowledge between programming language more obvious… (Case error, also the sentence is not clear).

Missing references:

• There are various approaches to solve clone detection task problem starting from algorithmic based methods (?) and continuing with modern machine learning ones ().

• The authors observe that the Mean Reciprocal Rank (MRR) scores of RoBERTa (?) and CodeBERT (?) for the code search task...

Some references are duplicated in the REFERENCE section.

---

### Official Review · Reviewer_2qh1 · 2022-10-22

**Confidence:** 4
**Correctness:** 4
**Technical Novelty And Significance:** 2
**Empirical Novelty And Significance:** 3
**Recommendation:** 3

**Clarity, Quality, Novelty And Reproducibility:**

Clarity:
- The current organization of the paper makes it a little difficult to follow. I would suggest restructuring. - Namely, the approach (Section 3) should be highlighted prior to describing datasets and evaluation metrics. A lot of low-level technical details (e.g., filtering, MRR formula) about prior work are presented early on, which take away from the focus of the contributions of this work. I would suggest moving these kinds of details to the appendix. Additionally, the tasks, datasets, and evaluation metrics are all mixed into the same sections right now, and I would suggest separating them out.
- Missing references with ? in many places.
- In Table 5, the defect-detection score is missing for one of the ablations, and the reason for excluding this is not clear.

Novelty:
The authors put forth a new dataset. The explicit task of cross-lingual clone detection is novel, though the more general idea of having language-agnostic representations of code in an embedding space have been studied before (https://arxiv.org/pdf/1806.07336.pdf). Additionally, contrastive learning has been used for pretraining in prior work (https://arxiv.org/pdf/2007.04973.pdf) for code clone detection. For these reasons, it seems that the novelty of this work is somewhat limited.

Quality:
The presentation of the paper could be improved (see suggestions in the “Clarity” section). Additionally, additional baselines and more experiments (see suggestions in “Weaknesses” section) would make this paper stronger.

Reproducibility:
There are a lot of missing details so it would be difficult to replicate these results.




**Strength And Weaknesses:**

Strengths:
- The idea of cross-lingual clone detection is nice, and it could be useful for acquiring parallel data for studying tasks like code translation.
- The analysis that shows that the model learns to embed similar programs of different programming languages near one another while other baselines tend to group by programming language is nice, though it is not clear whether the sample was random and the sample size is quite small.
- The ablation study in 5.1 is nice, as it breaks down the improvements achieved through the new data as well as the various loss components. This table shows that there is a clear advantage in using contrastive loss and combining it with the standard LM loss. Nonetheless, there are a few things that are clear about Table 5 (described below).

Weaknesses:
- There are many things that are unclear in the paper which make it difficult to understand some aspects of the technique as well as the experimental setup. Namely, the approach taken to compile XCD is not clear. It is written that there are three different setups: retrieval way, code clone detection way, and hybrid. There are not enough details given to understand these setups. Next, the size of the training set is not mentioned. Additionally, the zero-shot setting is not adequately described. The error detection loss is described as a binary classification task, but there are 8 classes described in 2.4.1, so it is not clear what the connection is. Table 5 suggests that there is some “verdict verification” component in the model, but this was never described in the paper. Also, in Table 5, it is not clear what “size limitation” is in reference to.
- The baselines used in this work are not at all described. Furthermore, it seems like authors could have considered stronger, more recent multilingual models of code like CodeGen.
- It seems that the authors could have also tested on some of the benchmarked code translation datasets, since they do provide pairs for cross-lingual clone detection.
- An ablation without the hard negative mining (either with a less stringent criteria for mining hard negatives or random sampling) would have been nice to further dive into the contrastive learning aspect of this work.
- Some analysis for in-domain vs out-of-domain cross-lingual clone detection would have made the paper stronger. For example, if you train on only languages A, B, and C, how well does the model generalize to detecting functionally equivalent programs in languages D and E.


**Summary Of The Paper:**

This paper presents an approach which incorporates contrastive learning into the training procedure to address the novel task of cross-lingual clone detection. For evaluation, authors construct a benchmark (XCD) by mining CodeForces. They show that their approach outperforms baselines on existing benchmarks for clone detection as well as on their new benchmark. On a sample of examples, they show that their model learns a latent embedding space in which cross-lingual examples implementing the same functionality are close together, in comparison to baselines which group examples of the same underlying programming language together.

Contributions:
- Formulation of the task of cross-lingual clone detection
- Training procedure which combines contrastive learning, standard LM loss, and error detection
- XCD benchmark for code clone detection which spans 9 different languages


**Summary Of The Review:**

Overall, there are a lot of missing details with respect to the techniques as well as the experimental settings, making it difficult to evaluate some aspects of this paper. While I appreciate the idea of explicitly studying cross-lingual clone detection, the general intuition has been previously studied before and the technique of using contrastive learning for clone detection has also been studied before, so I feel that the novelty of the work is somewhat limited.

---

### Official Review · Reviewer_Q8fk · 2022-10-24

**Confidence:** 5
**Correctness:** 2
**Technical Novelty And Significance:** 2
**Empirical Novelty And Significance:** Not applicable
**Recommendation:** 1

**Clarity, Quality, Novelty And Reproducibility:**

This paper should have been edited more rigorously before submission. It is incomplete in several places: multiple references resolve to `(?)` and others are missing entirely (e.g. in paragraph 2 of the introduction). The writing is of very low quality. Many (if not most) sentences are ungrammatical and the text often flows poorly, with sentences not connecting to the surrounding text. I would typically include a list of typos and ungrammatical sentences, but in this case that is unrealistic; please make several major editing passes before submitting this paper in the future.

Other than that, the main novelty lies in the created cross-lingual dataset. This dataset is relatively limited, focusing on many solutions but only to 100 distinct programs, and would be better suited for an empirical software engineering venue. As discussed above, the contrastive loss is not a significant contribution on its own. The `L_bug` loss _might_ be new, but its value is not studied in any ablations, as far as I can tell.

**Strength And Weaknesses:**

The work aims to show that cross-lingual training using a contrastive loss improves performance on several tasks that evaluate the semantics representing quality of code embeddings, and succeeds at doing so. However, the only novel aspect of this work is the multi-lingual dataset, which is a software engineering contribution (and also quite limited, considering just 100 distinct programming problems). The use of a contrastive loss is otherwise quite common place, even within "AI for Code" papers.

There are several other significant issues. For one, the heavy, almost exclusive use of GraphCodeBERT as a baseline is problematic because this model was not trained to perform well on this task. Its performance on cross-lingual code retrieval is effectively 0, which clearly highlights that stronger baselines should have been used. Many methods have been proposed for the tasks studied in this work, most of which could have been retrained or extended with the newly constructed cross-lingual dataset in order to provide much stronger baselines. On P8, the work mentions even mentions several candidate baseline methods for the code search task and states that these "[...] turned out to be still a strong baseline in our experiment", yet I could not find any of these in the results. This greatly undercuts the perceived usefulness of this technique.

Furthermore, the work suffers from major quality and clarity issues (see below). As a consequence, many experimental conditions are under- or unexplained. To name a few examples (there are too many to enumerate): Tab. 5 discusses a "size limitation" constraint that was never introduced in the methodology. Section 2.1 mentions MRR@20 but never indicates what the 20 refers to. The description for Eq. 1 (Sec. 3) mentions both languages and users, but neither appears in the equation. The fact that the model aims to discriminate between users also seems quite surprising given the stated goal of the work.

**Summary Of The Paper:**

This work constructs a multi-lingual dataset of solutions to programming problems in order to train a cross-lingual code clone detection model. This model is powered primarily by a contrastive loss that helps it learn lagnuage-agnostic (and instead problem-specific) representations of programs. The resulting model is evaluated on code clone, search & clone retrieval tasks where it generally achieves stronger performance than baselines.

**Summary Of The Review:**

The contributions in this work are limited and primarily geared towards a different audience. The writing needs to be substantially improved. Many experimental details are unclear or problematic. As a whole, this work is not fit for publication in its current form.

---

### Decision · Program_Chairs · 2023-01-20

**Decision:**

Reject

**Justification For Why Not Higher Score:**

Clear agreement from reviewers that the paper isn't ready, and no author response.

**Justification For Why Not Lower Score:**

N/A

**Metareview: Summary, Strengths And Weaknesses:**

Reviewers are in agreement that the paper has several significant flaws (writing, lack of novelty, unfair experimental setup) preventing it from being ready for publication, and there was no author response.